# From the Vector to Scalar Perturbations Addition in the Stark Broadening Theory of Spectral Lines

**Valery Astapenko** [1,*], **Andrei Letunov** [2,3] and **Valery Lisitsa** [1,2,3]

1 Moscow Institute of Physics and Technology (National Research University), Institutskij Per. 9, 141700 Dolgoprudnyj, Russia; vlisitsa@yandex.ru
2 National Research Centre "Kurchatov Institute", 123182 Moscow, Russia; letunovandrey11@yandex.ru
3 Institute for Laser and Plasma Technologies, National Research Nuclear University MEPhI, 115409 Moscow, Russia
* Correspondence: astval@mail.ru

**Abstract:** The effect of plasma Coulomb microfied dynamics on spectral line shapes is under consideration. The analytical solution of the problem is unachievable with famous Chandrasekhar–Von-Neumann results up to the present time. The alternative methods are connected with modeling of a real ion Coulomb field dynamics by approximate models. One of the most accurate theories of ions dynamics effect on line shapes in plasmas is the Frequency Fluctuation Model (FFM) tested by the comparison with plasma microfield numerical simulations. The goal of the present paper is to make a detailed comparison of the FFM results with analytical ones for the linear and quadratic Stark effects in different limiting cases. The main problem is connected with perturbation additions laws known to be vector for small particle velocities (static line shapes) and scalar for large velocities (the impact limit). The general solutions for line shapes known in the frame of scalar perturbation additions are used to test the FFM procedure. The difference between "scalar" and "vector" models is demonstrated both for linear and quadratic Stark effects. It is shown that correct transition from static to impact limits for linear Stark-effect needs in account of the dependence of electric field jumping frequency in FFM on the field strengths. However, the constant jumping frequency is quite satisfactory for description of the quadratic Stark-effect. The detailed numerical comparison for spectral line shapes in the frame of both scalar and vector perturbation additions with and without jumping frequency field dependence for the linear and quadratic Stark effects is presented.

**Keywords:** plasma spectroscopy; stark broadening; atomic physics





## 1. Introduction

The problem of the multiparticle perturbations effect from charged plasma particles on atomic spectra was recognized many years ago [1,2]. The fundamental solution for the field strength distribution function from an ensemble of static charged particles was obtained by Holtsmark [3]. The essential property of the Holtsmark function is its difference from the standard Gaussian distribution. This circumstance is connected with the vector nature of field strengths sum from individual particles, where the contribution of a specific member into the total sum is comparable with the total value. It is in contrast with the scalar sum of field modulus, where this contribution is statistically small. The dynamics of Coulomb field due to particles thermal motion can be recognized from Chandrasekhar–Von-Neumann analysis of particles (stars) motion with gravitational interaction [4]. Nevertheless, the direct transition from vector to scalar perturbation addition in the Stark broadening theory was not demonstrated analytically up to the present time.

The effect of a multiparticle electric field **E** on atomic spectra is characterized by the number of particles in the Weisskopf sphere

$$g = N \cdot r_W^3,$$ (1)

where $N$—the concentration of interacting particles and $r_W$ is the Weisskopf radius—the effective particle interaction radius.

The potential of binary interacting particles has the following form

$$V(r) = \frac{C_n}{r^n} \tag{2}$$

where $C_n$ is the interaction constant and $r$ is the distance between particles.

The parameter $g$ can be expressed in terms of $C_n$ and the density of particles. For the linear Stark effect, $g = N \cdot \left(\frac{C_2}{v_T}\right)^3$. As for the quadratic Stark effect, $g = N \cdot \frac{C_4}{v_T}$. Here, $v_T$ is the thermal velocity of the particles. When $g \gg 1$, one can use the statical approximation for line shapes (based on the vector perturbation sum), whereas for the case $g \ll 1$, the impact theory, based on scalar perturbation sum, is valid. Both cases are established in detail in the Stark broadening theory (see for example [5,6]). However, the correct theory for intermediate values of the parameter $g$ is absent. The first attempt to provide the solution to the problem of arbitrary $g$ for the linear Stark effect was undertaken by V.I. Kogan [7]. He expressed their results for the spectral line profile in terms of complicated path integrals. Unfortunately, he managed to find only the thermal corrections to the statical Holtsmark profile.

A general solution for the quadratic Stark effect was given in the famous paper of H.R. Griem, M. Baranger, A.C. Kolb and G. Oertel (GBKO) [1]. Their result is based on the scalar addition of perturbation following the general scalar theory from older papers on the broadening theory (see [2]). The general GKBO expression for the intensity profile reproduces the impact limit for high velocities of ions. However, when coming to the specific calculations for $g \gg 1$ in helium spectra, the authors of [1] used the "vector" static Holtsmark distribution function. Note that the scalar theory is not automatically equal to the binary one as it follows from general results for the Van der Waals interaction ($n = 6$ in Equation (2)). This problem was considered by Chen and Takeo (CT) in [2].

In 1971, Brissaud and Frisch gave an impetus to the problem of the Stark broadening. They introduced a new approach [8] of the spectral line shape calculation called the Model Microfield Method (MMM). This theory is based on the assumption that all fluctuations of electric microfield could be treated as a Markovian process. This model has been widely used in the theory of spectral line broadening [9–11]. Better agreement with the results of modeling the formation of the spectral line shape yielded the Frequency Fluctuation Model (FFM) [12]. The main idea of this theory is that the electric field fluctuations induce fluctuations of the radiation intensity and of the radiation frequency. This approach was applied to many problems of spectral line calculations in plasma (see for example [13–16]). In the work [17], it was shown that the usage of the FFM and the method of the kinetic equation are equivalent. Both of these approaches lead to the same result: the resulting spectral line shape is the functional of the static profile.

Molecular dynamics methods and computer simulations make it possible to perform the most accurate calculations of a spectral line shape in plasmas (see for example [18–20]). However, with a focus on accuracy, it is necessary to increase the number of particles in the simulation, which requires the use of more computational resources. The FFM is believed to provide the most accurate description of a spectral line profile under the action of moving ions and lets one perform fast calculation of the intensity profile [21]. However, in the paper [14], it was pointed out that the FFM does not reproduce the correct behavior of the profile width in the impact limit for hydrogenic plasma. In the present paper, we will show how the FFM can be modified to yield the correct behavior of the profile in the impact limit. Moreover, we will compare the FFM profiles with the expression obtained in [1] for the quadratic Stark effect.

## 2. The Goal of the Paper

The main purpose of this article is to test the FFM procedure. If the FFM describes the influence of ionic thermal motion on the spectral line shape with a good accuracy, it means that the usage of the FFM can replace complicated methods of molecular dynamics and complex analytical solutions like GKBO [1] (which, in fact, does not work for low temperatures). In the present paper, we will consider two cases: the linear and quadratic Stark effects.

For the linear Stark effect, there is a problem that is connected with the incorrect dependence of the impact width on the ion's velocity [14]. In the present paper, we will introduce the modification of the FFM procedure, which consists of replacing the constant jumping frequency with the non-constant one obtained in the paper [4]. Our goal is to show that this action leads to the correct behavior of the intensity profile width in the impact limit. In order to do this, we will perform analytical and numerical calculations.

The estimations of the impact width in the case of the quadratic Stark effect is one of the goals of the present paper, too. Moreover, we will make the detailed comparison of the FFM with the results of the GKBO theory.

## 3. A Brief Description of the FFM

The FFM procedure was suggested in order to overcome the complicated dynamics of a Coulomb ion's microfield [12]. At the first step, the FFM procedure was based on the idea that ion's thermal motion resulting in the field fluctuation would make jumps between different Stark components with a frequency, which is equal to

$$\nu = N_i^{1/3} v_{Ti},\tag{3}$$

where $N_i$ is the density and $v_{Ti}$ is the thermal velocity of ions in plasma.

The corresponding procedure was entirely numerical and demonstrated a great success in line shapes description tested by the comparison with computer codes based on molecular dynamics (MMD) [19,21]. The essential development of the FFM was the theorem about equivalence of the FFM procedure to the solution of the kinetic equation with so-called "the strong collision integral" describing just the dynamics of ion microfield in terms of jumps between different values of the field strengths responsible for time evolution of Stark components [17]. The presentation of the FFM in terms of the kinetic equation made it possible to obtain an analytical solution for the FFM spectral line shapes. The solution contains only the electric field distribution function (usually Holtsmark) and the jumping frequency so the total FFM shape was the functional of the static field distribution function. The resulting intensity profile $I(\omega)$ can be expressed as the functional of the normalized statistical profile $W(\omega)$ (see [17])

$$I(\omega) = \frac{1}{\pi} Re \frac{\int \frac{W(\omega')d\omega'}{\nu + i(\omega - \omega')}}{1 - \nu \int \frac{W(\omega')d\omega'}{\nu + i(\omega - \omega')}},\tag{4}$$

where $\omega$ is the energy shift from unperturbed spectral line[1].

Equation (4) can be rewritten in another form that is more convenient for calculations:

$$I(\omega) = \frac{\nu}{\pi} \frac{J_0(\omega)J_2(\omega) - J_1^2(\omega))}{J_2^2(\omega) + \nu^2 J_1^2(\omega)},\tag{5}$$

where

$$J_k(\omega) = \int\limits_{-\infty}^{+\infty} \frac{W(\omega')(\omega - \omega')^k d\omega'}{\nu^2 + (\omega - \omega')^2}.\tag{6}$$

There is the connection between $J_0(\omega)$ and $J_2(\omega)$, which might help one to perform calculations of the spectral line shape faster

$$J_2(\omega) = 1 - \nu^2 J_0(\omega). \tag{7}$$

When $\nu \to 0$ Equation (4) turns into the statistical limit. In the denominator of (4) we can neglect the term that is proportional to $\nu$ and obtain the following expression:

$$I(\omega) = \frac{1}{\pi} Re \int \frac{W(\omega')(\nu - i(\omega - \omega'))d\omega'}{\nu^2 + (\omega - \omega')^2} = \int W(\omega')\delta(\omega' - \omega)d\omega' = W(\omega) \tag{8}$$

The characteristic value of energy shift of the considered Stark component is equal to

$$\Omega_n^S = C_n^S N_i^{\frac{n}{3}}, \tag{9}$$

where $C_n^S$ is the constant of the Stark effect, which depends on the atomic quantum states.

For the linear Stark effect $n = 2$ in Equation (9). In a non-hydrogenic plasma the ionic broadening is determined by the quadratic Stark effect and $n = 4$.

In the present paper, we will restrict ourselves to considering a single Stark component. Different methods of approximation of the array of radiative transitions are presented in [22–24]. The considered spectral profile can be treated as a component with an averaged Stark shift and intensity (see [5]). Note that in the case of the linear Stark effect, every spectral component has absolutely the same pair, but with negative value of the energy shift (9). It is convenient to use the reduced energy shift, which is determined by the following relation:

$$z = \frac{\omega}{\Omega_n^S} \tag{10}$$

Equations (5) and (6) will turn into the following formulas

$$I(z) = \frac{\bar{\nu}}{\pi} \frac{J_0(z)J_2(z) - J_1^2(z))}{J_2^2(z) + \bar{\nu}^2 J_1^2(z)}, \tag{11}$$

$$J_k(z) = \int\limits_{-\infty}^{+\infty} \frac{W(z')(z - z')^k dz'}{\bar{\nu}^2 + (z - z')^2}, \tag{12}$$

where

$$\bar{\nu} = \frac{\nu}{\Omega_n^S}. \tag{13}$$

The relations (7) and (8) remain the same in the framework of the reduced variables.

In the present paper, we will use a non-constant jumping frequency. In order to modify the FFM, the jumping frequency will be considered as the function of the energy shift $\bar{\nu} \to f(\bar{\nu}, z)$. In this case, it is necessary to make the modification of Equations (11) and (12):

$$I(z) = \frac{1}{\pi} \frac{J_0'(z)J_2(z) - J_1(z)J_1'(z)}{J_2^2(z) + J_1'^2(z)}, \tag{14}$$

where

$$J_k(z) = \int\limits_{-\infty}^{+\infty} \frac{W(z')(z - z')^k dz'}{f^2(\bar{\nu}, z') + (z - z')^2}, \tag{15}$$

$$J_k'(z) = \int\limits_{-\infty}^{+\infty} \frac{f(\bar{\nu}, z')W(z')(z - z')^k dz'}{f^2(\bar{\nu}, z') + (z - z')^2}, \tag{16}$$

It is easy to see that in the case of the constant jumping frequency $f(\bar{\nu}, z) = \bar{\nu}$, Equations (14)–(16) turn into Equations (11) and (12).

Equations (11)–(16) present a general solution of the spectral line broadening problem, describing a smooth transition from static to the impact limit. This method is applicable for arbitrary values of the parameter (1) including $g = 1$.

## 4. Stark Broadening of Hydrogen Spectral Lines

In the case of the Holtsmark theory, the static profile is equal to the following Equation ([3,5]):

$$W(z) = \frac{1}{k} H\left(\frac{z}{k}\right), \tag{17}$$

where $k = 2\pi\left(\dfrac{4}{15}\right)^{\frac{2}{3}} \approx 2.6031$ and $H(x)$ is the Holtsmark function, which is equal to

$$H(x) = \frac{2}{\pi x} \int_0^\infty t \sin t \exp\left[-\left(\frac{t}{x}\right)^{\frac{3}{2}}\right] dt. \tag{18}$$

In the impact limit, when a velocity of particles is high ($v \to \infty$), the spectral line shape must transform into the Lorentz profile (see for example [5]):

$$L(z) = \frac{1}{\pi} \frac{\gamma}{\gamma^2 + (z - z_0)^2}, \tag{19}$$

where $\gamma$ is the width of the profile and $z_0$ is the coordinate of the center of the Lorentz profile.

In the case of the linear Stark effect $\gamma \sim \dfrac{1}{v}$ [5]. However, in the work [14], the authors showed that the joint usage of the FFM and the quasi-contiguous approximation for the line shape modeling in plasmas [22] leads to the wrong result in the impact limit: $\gamma_{FFM} \sim \dfrac{1}{\sqrt{v}}$. We shall show how to reach the correct result for the line width by introducing a non-constant jumping frequency.

In order to give a solution to this problem and overcome this discrepancy, we will modify the FFM by introducing the non-constant jumping frequency $f(v, z)$. To perform calculation of the spectral line shape of the single Stark component, we will use Equation (14). The field dependent jumping frequency can be extracted from the paper [4] by interpolation between small and large values of an electric field strength[2]

$$f_c(\bar{v}, z) = \bar{v} \frac{3.67}{\tau(z)}, \tag{20}$$

where

$$\tau(z) = \frac{1.37z}{1.37 + z^{\frac{3}{2}}}. \tag{21}$$

While calculating the spectral line shape in a plasma one deals with pairs of the same Stark components, which are symmetric about the center of the spectral line. In the case of a single Stark component, integrands in (16) are actually integrated from zero to infinity. Consideration of two Stark components implies integration from minus to plus infinity. In order to consider this case, we need to change the static profile $W(z)$ to $W(|z|)$.

We can perform simple analytical estimations of the impact width. In order to do this, we will use the fact that all normalized profiles $I(z)$, which are symmetric around zero, have the following property:

$$I(0) \sim \frac{1}{\gamma}, \tag{22}$$

where $\gamma$ is the width of the profile $I(z)$.

Firstly, we will obtain the dependence on the impact width in the case of the constant jumping frequency. Using Equation (12), it is easy to see that $J_1(0) = 0$ because we integrate the odd function over the symmetric interval. Using Equation (11), one will obtain

$$I(0) = \frac{\bar{v}}{\pi} \frac{J_0(0)}{J_2(0)}.\tag{23}$$

Now we need to estimate the values of $J_0(0)$ and $J_2(0)$ when $\bar{v} \gg 1$.

$$J_0(0) = \frac{2}{\bar{v}^2} \int\limits_0^{+\infty} \frac{W(z')dz'}{1 + \frac{(z-z')^2}{\bar{v}^2}} \sim \int\limits_{\bar{v}}^{\infty} \frac{W(z')dz'}{1 + \frac{(z-z')^2}{\bar{v}^2}} \sim \frac{2}{\bar{v}^2} + \frac{const_1}{\bar{v}^{7/2}}\tag{24}$$

$$J_2(0) = \frac{2}{\bar{v}^2} \int\limits_0^{+\infty} \frac{W(z')z'^2dz'}{1 + \frac{(z-z')^2}{\bar{v}^2}} \sim \frac{const_2}{\bar{v}^{3/2}}\tag{25}$$

In these calculations, we used the asymptotic behavior of the Holtsmark profile $W(z \gg 1) \sim z^{-5/2}$. While calculating these integrals, it should be taken into account that the main contribution to the value of the integral is made by the interval from $\bar{v}$ to $\infty$.

Substitution of (24) and (25) in Equation (23) leads to the following result:

$$I(0) \sim \frac{1}{\bar{v}^{1/2}}.\tag{26}$$

After comparison of Equations (26) and (22), we can conclude that for large values of $\bar{v}$: $\gamma_{FFM} \sim \frac{1}{\bar{v}^{1/2}}$. This dependence of the impact width on the parameter $\bar{v}$, obtained for the two symmetrical Stark components, coincides with the result from [14].

In the case of the non-constant jumping frequency, we also have to change $f_c(\bar{v}, z)$ to $f_c(\bar{v}, |z|)$. According to Equation (15)[3], the value of $J_1(0)$ is much smaller than all other FFM functions when $z = 0$. Because of that, we can neglect the second terms both in numerator and denominator of Equation (14). It leads to the following formula for the non-constant jumping frequency:

$$I(0) = \frac{1}{\pi} \frac{J_0'(0)}{J_2(0)}.\tag{27}$$

Using again the asymptotic behavior of the Holtsmark profile and the formula for Chandrasekhar—von Neumann jumping frequency (20), we will estimate the values of $J_0'(0)$ and $J_2(0)$.

$$J_0'(0) \sim \frac{1}{\bar{v}},\tag{28}$$

$$J_2(0) \sim \frac{1}{\bar{v}^2},\tag{29}$$

$$I(0) \sim \frac{1}{\bar{v}}.\tag{30}$$

Equations (27)–(30) lead to the following dependence of the impact width on $\bar{v}$: $\gamma_{FFM} \sim \frac{1}{\bar{v}}$. According to the basic results of the impact theory [5], this is correct behavior of the profile width.

The normalized intensity profiles as the function of the reduced energy shift is presented in Figure 1. Using Equation (14), we performed calculations for the spectral line shape of two symmetrical Stark components in two cases: $f(\bar{v}, z) = \bar{v}$ (constant) and $f(\bar{v}, z) = f_c(\bar{v}, z)$ according to Equation (20) (non-constant). For small ion velocities or high densities, according to the relation (8), the intensity profiles are close to each other. As the parameter $\bar{v}$ grows, the difference between graphs increases. For large values of $\bar{v}$, width of

profiles calculated using non-constant jumping frequency is noticeably lower than in the case of constant $f(\bar{v}, z) = \bar{v}$.

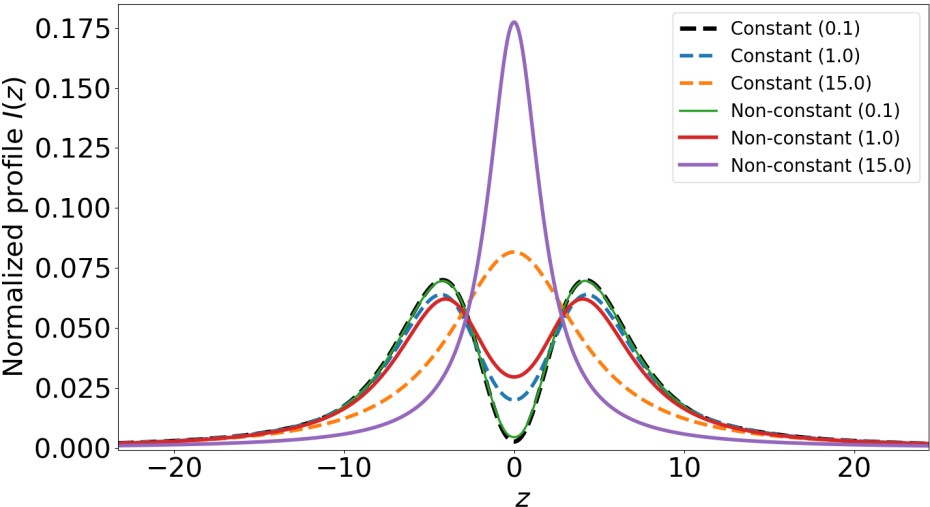

**Figure 1.** The normalized intensity profile of two symmetrical Stark components as the function of the reduced energy shift. Comparison of the FFM profiles calculated with the constant and non-constant jumping frequencies. Calculations are presented for different values of ion's velocity. The value of the reduced constant jumping frequency $\bar{v}$ is written in the brackets.

Comparison of the FFM profiles in the cases of constant and non-constant jumping frequencies with it's approximation by the function (19) is presented in Figure 2. In this calculation, we put $\bar{v} = 15$. In order to do this approximation, we used the the method of least squares. It is easy to see that the FFM reproduces a Lorentzian profile in the impact limit. However, the width of the profiles in the cases of the constant jumping frequency $f(\bar{v}, z) = \bar{v}$ and non-constant $f(\bar{v}, z) = f_c(\bar{v}, z)$ have completely different values. One of the goals of the present paper is to confirm the analytical results (26) and (30) for the impact width $\gamma = \gamma_{FFM}$ by numerical calculations. In order to do that, we will calculate the FFM profiles with different values of $\bar{v}$. Then every intensity profile will be approximated by the Lorentzian (19). After this fitting, we will approximate the obtained impact width by the following functions:

$$y_c = \frac{A}{\sqrt{\bar{v}}}, \tag{31}$$

for the constant jumping frequency

$$y_n = \frac{B}{\bar{v}}, \tag{32}$$

for the non-constant jumping frequency.

In Figure 3, one can see the values of the impact width as the function of the constant jumping frequency $\bar{v}^4$. The specific curves (31) and (32) that approximate the function $\gamma_{FFM} = \gamma_{FFM}(\bar{v})$ are also presented in Figure 3. We used the method of least squares to approximate the function $\gamma_{FFM}(\bar{v})$.

Figure 3 shows that the analytical results for the impact width are in agreement with numerical calculations.

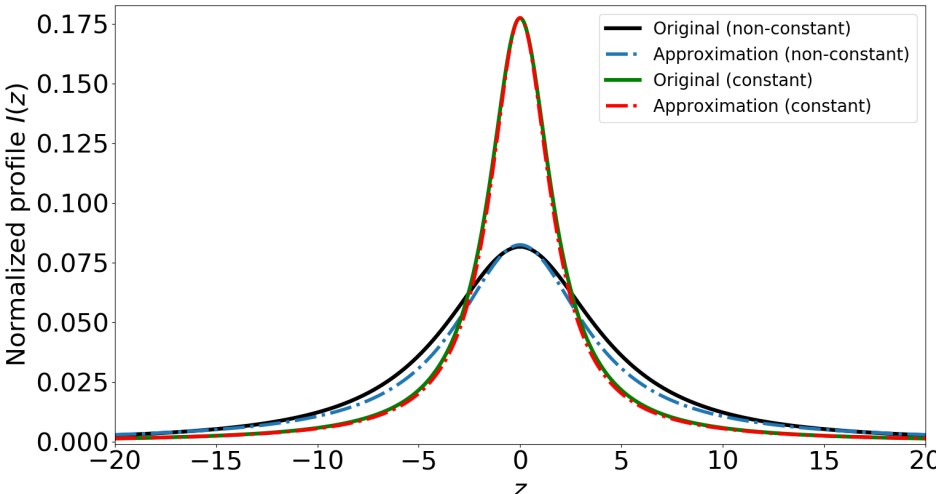

**Figure 2.** The normalized intensity profile of two symmetrical Stark components as the function of the reduced energy shift. Comparison of the FFM profiles calculated with constant and non-constant jumping frequency and its approximation by the Lorentz profile; $\bar{v} = 15$.

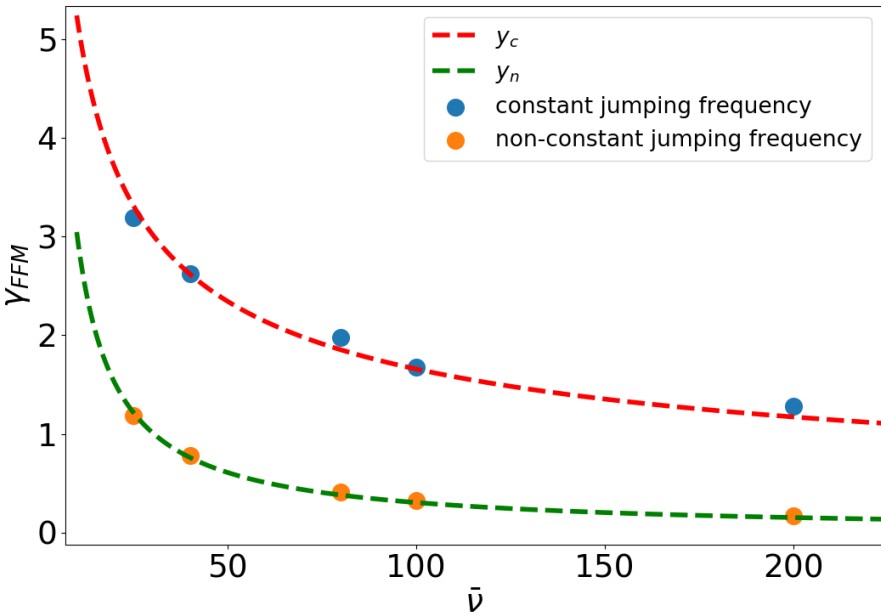

**Figure 3.** The impact width as the function of the constant jumping frequency $\bar{v}$. The comparison of two cases of spectral line shape calculation: the usage of constant and non-constant jumping frequency; furthermore, presented the specific curves $y_c = \dfrac{16.6}{\sqrt{\bar{v}}}$ and $y_n = \dfrac{30.4}{\bar{v}}$, which approximate the dependence of the impact width on $\bar{v}$.

Within the margin of numerical error (double usage of the least squares method), we can conclude that $\gamma_{FFM}$ can be approximated by Equations (31) and (32) with a good accuracy.

To sum it all up, the modification of the FFM, which consists in the usage of the non-constant jumping frequency, yields the correct results for the limit of low and high temperatures. For the case of $\bar{v} \lesssim 1$, the profiles calculated with constant and non-constant jumping frequency are close to each other. The results presented in [19,21] show that when $\bar{v} \sim 1$, the FFM calculations are in agreement with computer modeling. So we can conclude that the modified FFM gives the correct description of the spectral line shape for all temperatures.

## 5. Stark Broadening of Non-Hydrogenic Spectral Lines

In the case of non-hydrogenic atoms, the ionic broadening is determined by the quadratic Stark effect. It leads to the transformation of the static profile $W(z)$

$$W_q(z) = \frac{1}{2\sqrt{z}}W(\sqrt{z}), \tag{33}$$

where $W_q^v(z)$ is the statistical profile of the single Stark component for the energy shift, which is proportional to the squared electric filed.

Equation (17) will take the form

$$W_q^v(z) = \frac{1}{2\sqrt{z}k}H\left(\frac{\sqrt{z}}{k}\right), \tag{34}$$

where the upper index 'v' means that this profile was obtained using the vector addition theory.

Using the theory of the scalar addition of perturbations, the authors in works [1] (GKBO) and [2] (CT) obtained the static profile in the case of the quadratic Stark effect:

$$W_q^s(z) = C^s \cdot \int_0^\infty \cos\left(zt - at^{\frac{3}{4}}\right)\exp\left(-bt^{\frac{3}{4}}\right)dt, \tag{35}$$

where $a = 14.0309$ and $b = 5.81178$;

$$C^s = \left[\int_{-\infty}^{+\infty}dz\int_0^\infty \cos\left(zt - at^{\frac{3}{4}}\right)\exp\left(-bt^{\frac{3}{4}}\right)dt\right]^{-1} \tag{36}$$

The scalar addition theory is not applicable in the static case $\bar{v} \longrightarrow 1$. It means that Equation (35) is not correct. However, it would be interesting to compare the Holtsmark profile (34) with (35) and understand how scalar and vector theories differ from each other.

Figure 4 shows graphs of Equations (34) and (35). For small values of $z$, one can see a big difference between two profiles. The scalar one is equal to zero for $z < 8.1$. This strange result is connected with strong oscillations of the integrand in Equation (35). Both of these functions decrease much slower than the Holtsmark static profile for the linear Stark effect.

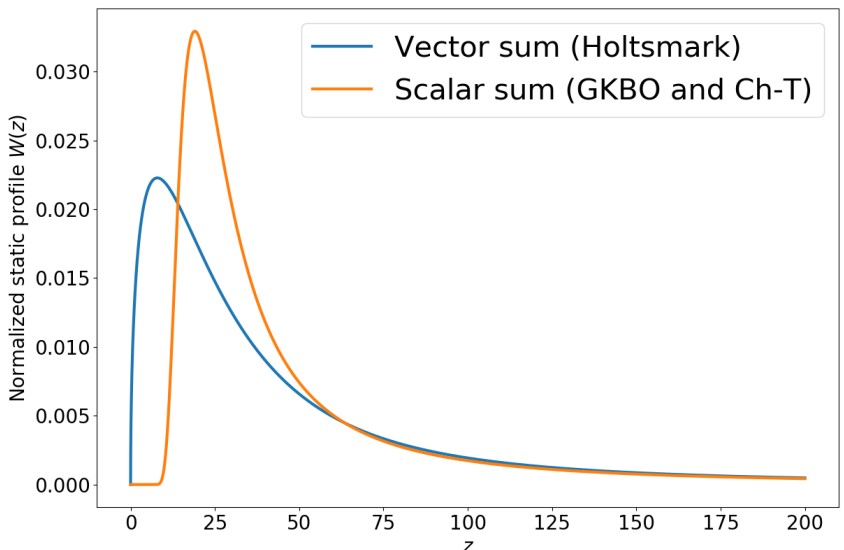

**Figure 4.** The normalized static profile as the function of the reduced energy shift. The comparison of two cases: vector (Holtsmark) and scalar (CT and GKBO) perturbation addition theories.

In the work [2], the authors obtained the general expression for a static profile for any type of interaction (in the framework of the theory of scalar perturbation addition). However, for the linear Stark effect the formula for the static profile is expressed in terms of the divergent integrals. For this reason, we do not provide a comparison similar to shown in Figure 4.

Using the theory of scalar addition of perturbations, the authors of [1] obtained the intensity profile (GKBO) of a single component for any ion temperature:

$$I_{GKBO}(\bar{v}, z) = \frac{1}{\pi} Re\left[ \int\limits_0^\infty dt \exp\left( izt + g(\bar{v}, t)\right)\right], \tag{37}$$

where

$$g(\bar{v}, t) = 2\pi\bar{v}\int\limits_0^\infty \rho d\rho \int\limits_{-\infty}^{+\infty} dt_1\left( \exp\left\{ -i\frac{1}{2\bar{v}\rho^3}\varphi(\bar{v}, t, t_1, \rho)\right\} - 1\right), \tag{38}$$

$$\varphi(\bar{v}, t, t_1, \rho) = \arctan\left\{ \frac{\bar{v}(t - t_1)}{\rho}\right\} + \arctan\left\{ \frac{\bar{v}t_1}{\rho}\right\} + \frac{\bar{v}(t - t_1)\rho}{\rho^2 + \bar{v}^2(t - t_1)^2} + \frac{\bar{v}t_1\rho}{\rho^2 + \bar{v}^2 t_1^2} \tag{39}$$

For large values of $\bar{v}$, Equation (37) reproduces the impact theory for the quadratic Stark effect:

$$I_{GKBO}(\bar{v} \gg 1, z) = \frac{1}{\pi}\frac{\gamma_{GKBO}}{\gamma_{GKBO}^2 + (z - \Delta_{GKBO})^2}, \tag{40}$$

where

$$\gamma_{GKBO} = \bar{v}^{\frac{1}{3}}\pi\left(\frac{\pi}{2}\right)^{\frac{2}{3}}\Gamma\left(\frac{1}{3}\right)\cos\frac{\pi}{3} = 5.6863\bar{v}^{\frac{1}{3}}, \tag{41}$$

$$\Delta_{GKBO} = \bar{v}^{\frac{1}{3}}\pi\left(\frac{\pi}{2}\right)^{\frac{2}{3}}\Gamma\left(\frac{1}{3}\right)\sin\frac{\pi}{3} = 9.8489\bar{v}^{\frac{1}{3}}, \tag{42}$$

where $\Gamma(z)$ is the gamma function.

The comparison of the FFM and GKBO profiles for $\bar{v} = 25$ is presented in the Figure 5. These calculations show that the usage of the constant jumping frequency works much better than non-constant. The closest to GKBO graph is the FFM profile with constant frequency and Holtsmark static profile (34).

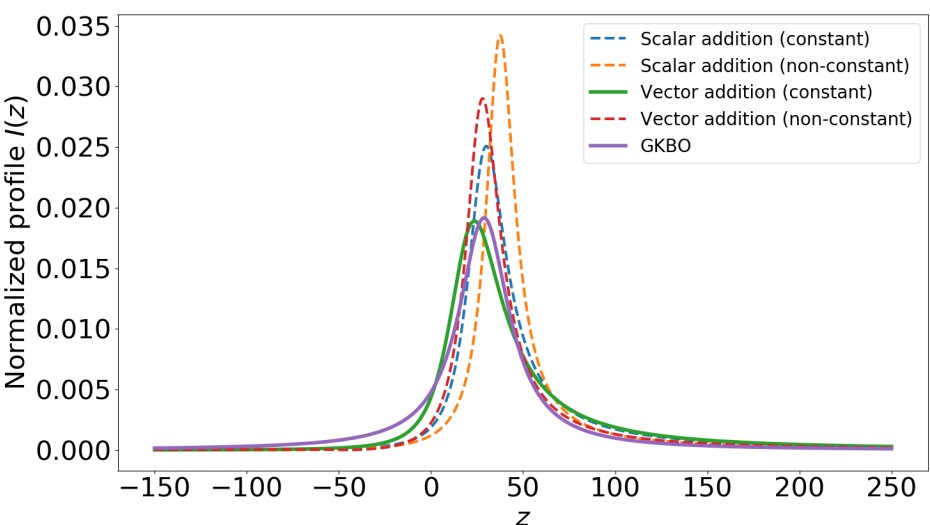

**Figure 5.** The normalized intensity profile as the function of the reduced energy shift. The comparison of the FFM profiles calculated using the theories of scalar and vector addition as well as constant and non-constant jumping frequencies with GKBO profiles; $\bar{v} = 25$.

The comparison of the FFM and GKBO profiles is presented in Figure 6. Firstly, the usage of the Holtsmark distribution (34) as the static profile in the FFM procedure gives better results for high temperatures than the scalar one (35). Secondly, when $\bar{v} \lesssim 1$, the shape of the FFM profiles changes very slowly. Finally, the FFM spectral line shapes for the quadratic Stark effect are asymmetric about the center of the profile. As we can see from Figures 5 and 6, for $\bar{v} \gg 1$, the joint usage of the FFM and the theory of vector addition of perturbations gives the results which are close to the GKBO profiles.

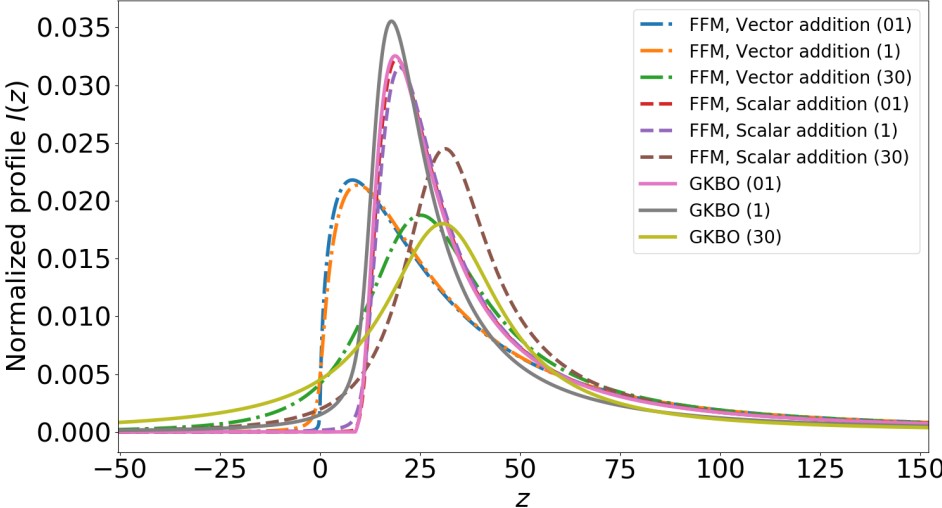

**Figure 6.** The normalized intensity profile as the function of the reduced energy shift. The comparison of the FFM profiles calculated using the theories of scalar and vector addition with GKBO profiles; values of $\bar{v}$ are written in the brackets.

The asymptotic estimations for $\bar{v} \gg 1$, similar to those that were done for the Linear Stark effect, of the integrals (13) leads to the following results:

$$\begin{cases} J_0(0) \sim \dfrac{1}{\bar{v}^2}, \\ J_1(0) \sim \dfrac{1}{\bar{v}^{7/4}}, \\ J_2(0) \sim \dfrac{1}{\bar{v}^{3/4}}. \end{cases} \tag{43}$$

It is easy to see that in the framework of asymptotics (43), the second terms both in numerator and denominator of (11) could be neglected. Again, we can use the relation (22). In this case, $\gamma_{FFM} \sim \bar{v}^{1/4}$, which is slightly different from the correct GKBO result. As numerical calculations show this "impact" asymptotic starts working for $\bar{v} \sim 10^4$. This slow convergence to the impact asymptotic is connected with the fact that now the FFM profile center is not located at zero. These estimations work when we can neglect the shift of the profile compared to its width. However, for lower $\bar{v}$, the coordinate of the maximum of the profile is of the same order of magnitude as it's width. Such values of $\bar{v}$ are non-realistic for ions in plasmas. So, this asymptotic of the impact width should be treated just as mathematical fact.

Direct use of Equation (37) will make calculations of a spectral line shape very cumbersome. Moreover, the GKBO results are incorrect for low temperatures (high densities). The FFM allows one to perform calculations of a spectral line shape much faster and it gives correct results for low temperatures (high densities). The numerical calculations, analogous to those which were done for the linear Stark effect, shows that in the impact limit, the profile width changes slowly. This means that the FFM produce the impact limit for the quadratic Stark effect ($\gamma \sim \bar{v}^{1/3}$). The verified analytical theory for the case of $\bar{v} \sim 1$ does not exist. However, in the work [21], it was shown that the FFM spectral line shape

is close to the profile obtained by the usage of method of molecular dynamics for $\bar{v} \sim 1$. We can conclude that the FFM can be a good approximation for the intensity profile of a Stark component when $0 < \bar{v} < 30$.

## 6. Conclusions

The problem of the influence of ions' thermal motion on the shape of spectral lines in plasmas has not been solved yet. There is the strict analytical solution for the case of zero temperature of ions. This result is obtained in the framework of the theory of the vector addition of perturbations from every particle. Furthermore, V.I. Kogan obtained the thermal corrections to this result [7]. However, if we change vector addition of perturbations to scalar [1]), we will obtain the result, which is correct for high temperatures. Unfortunately, the strict theory for intermediate values of temperature has not been constructed yet.

Consideration of the ionic motion as a stochastic process leads us to the FFM procedure. The main results of the FFM is that spectral line shape in a plasma with arbitrary temperature depends on the static profile and the jumping frequency (3) (see Equation (4)). It allows one to perform fast calculations of the intensity profile.

In the impact limit, the FFM procedure with the constant jumping frequency gives the wrong dependence of the impact width on the ion's thermal velocity in the case of the linear Stark effect (see the work [14] and Figure 3). However, if we change the constant jumping frequency to Equation (20), obtained in the work [4], the FFM will reproduce the impact limit. Thus, we can strictly assert that the modified FFM gives the correct results in the limits of low and high temperatures. The numerical analysis and comparison with computer modeling [21] of an ion's microfield dynamics shows that the FFM works well in the case of $\bar{v} \sim 1$ (see Equation (13)). We can conclude that the modified FFM is applicable for any ionic temperatures and densities.

For the quadratic Stark effect, the FFM procedure does not turn the static profile to the impact one. Analytical estimations show that for large values of $\bar{v}$, the behavior of the impact width is close to the correct one, but does not exactly match ($\gamma_{GKBO}/\gamma_{FFM} \sim \bar{v}^{1/12}$). Note that the convergence of the FFM profile to the impact asymptotic is very slow. It is achieved at non-physical values of $\bar{v}$. However, for high ionic temperatures, the joint usage of the FFM with constant jumping frequency and the Holtsmark theory (vector addition) yields the results that are very close to the GKBO profiles. The numerical analysis demonstrated in the present paper and the results from the work [21] show that the FFM approximates the spectral line shape of a Stark broadened profile in the physical domain $0 < \bar{v} < 30$.

**Author Contributions:** Conceptualization, V.L., V.A. and A.L.; methodology, V.L. and V.A.; software, A.L.; validation, V.L., V.A. and A.L.; formal analysis, V.L.; investigation, V.L., V.A. and A.L.; resources, A.L.; data curation, V.L.; writing—original draft preparation, A.L.; writing—review and editing, V.L.; visualization, A.L.; supervision, V.L. and V.A.; project administration, V.L. All authors have read and agreed to the published version of the manuscript.

**Funding:** This research received no external funding.

**Institutional Review Board Statement:** Not applicable.

**Informed Consent Statement:** Not applicable.

**Conflicts of Interest:** The authors declare no conflict of interest.

## Abbreviations

The following abbreviations are used in this manuscript:

| | |
|---|---|
| FFM | Frequency fluctuation model |
| GKBO | Griem, Baranger, Kolb and Oertel |
| MMM | Model Microfield Method |
| CT | Chen and Takeo |

## Notes

1　All over the paper atomic units are used: $e = m = \hbar = 1$, where $e$— the elementary charge, $m$—the mass of an electron, $\hbar$ is the Planck constant.

2　More details about stochastic problems connected with field distribution one can find in [25].

3　This fact was verified by numerical calculations.

4　Situation when $\bar{v} \gg 10$ rarely realized for ions in practice. Realistic values of $\bar{v}$ one can find in [5]. Calculations for such big $\bar{v}$ were performed to show how the FFM formally turns into the impact theory.

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
