# Peer review of "From the Vector to Scalar Perturbations Addition in the Stark Broadening Theory of Spectral Lines"

_universe, doi:10.3390/universe7060176_

Round 1
Reviewer 1 Report
The manuscript discusses the idea of using a frequency-dependent fluctuation frequency in the context of the FFM in order to fix a basic FFM flaw, i.e the impact limit, which the FFM cannot handle. The manuscript in its present from is very far from being acceptable because:
-it reiterates a number of older and false or misleading assertions, such as the Weisskopf radius falacy
-it makes a number of false statements such as (line 74)” The FFM is believed to provide the most accurate description of spectral line profile under the action of moving ions”. This is not true and even the authors in the same manuscript acknowledge that the FFM does not work in the impact regime.
-it neglects important literature. For instance it was well- known and and understood that the FFM does not recover the impact limit. Ways to remedy this have also been proposed and illustrated , see for instance High Energy Density Physics 9 (2013) p375, and particularly Fig.6 where those particles that are impact are treated as such (and NOT necessarily by perturbation theory, as in say GBKO) and the rest are treated via the FFM. In fact the impact limit critically depends on a number of details such as unitarity(see for instance Phys. Rev. Lett. 75, 3406 (1995) , perturbing level spacing, penetrating collisions(see for instance High Energy Density Physics 35, 100743 (2020), Physical Review E 72(4 Pt 2):046404 (2005) and -at least for electrons- quantum effects such as resonances, see for instance Phys. Rev. Lett. 124, 055003(2020) and references within. So the authors are not the first to propose how to fix the FFM in this respect.
-It completely confuses the reader, by mentioning that the FFM agrees with MD, then uses the Holtsmark distribution in Section 4 which not only neglects ion correlations, but uses unshielded field. It calls GBKO “complex analytical solutions”. Not sure why GBKO is “complex”, when it uses perturbation theory and pretending that complexity does not matter is not true if you want a general method.
-the English is very poor
In detail:
-Lines 21-33 should be rewritten as follows:
The effect of the plasma particles on atomic spectra has received a lot of attention for a long time, see for instance [1,2]. When these particles are static, the key quantity is their field strength distribution, first studied in the simplest approximation by Holtsmark[3]. This distribution is not Gaussian and this fact is related to the vector nature of the individual fields, where the contribution of typical fields is comparable to the total field strength, in contrast to scalar sums, where this contribution is vanishingly small. The importance of thermal particle motion bears analogies with the Chandrasekhar-Von Neuman analysis of the motion of stars in the context of a gravitational rather than Coulomb interaction[4].
-I do not understand the last sentence (lines 32-33) of the first paragraph. What transition?
-Get rid of the discussion on Weisskopf radius(up to line 57)-but use the definition of particle density N (call it particle density, not concentration, to avoid confusion). As was first pointed out by Seidel in the 1980 Spectral Line Shapes conference, the Weisskopf radius is a measure of unitarity and strength of the interaction. Strong collisions do not mean static-this is a common fallacy that should not be repeated. As has also been pointed out repeatedly in the literature, it leads to the absurd result that at high enough densities even electrons would be static, in which case for a line with a central component there is nothing in terms of pressure broadening to avoid a delta-function. The author’s assertion that for g>>1 one can use the static (not statical) approximation is simply wrong, as illustrated by the above example. This approach is very old and has no place today where ion dynamics is understood. The GBKO reference makes no sense to me and no point. GBKO used the impact theory for electrons and quasistatic for ions. We are in 2021 and references to Chen and Takeo with respect to scalar theory are hopelessly outdated. Chen and Takeo were trying to unify the Margenau wing theory with scalar impact theory, both of which have been supersεeded. Nowadays the term ‘scalar theory’ (JQSRT Vol 13. pp 1523-1538(1973))” specifically refers to neglecting time ordering. GBKO uses lowest order perturbation theory, so this is not a problem.
-Line 69: replace ‘the functional’ by ‘a functional’
-Lines 74-80: Replace “The FFM is believed ..profile[20] In fact replace lines 74-80 as follows:
“The FFM[20] is a very fast method that accounts for ions dynamics and yields accurate results, as long as we are far from the ion impact regime, where it fails[13]. Ways to remedy this have been proposed, e.g. [-here cite what has been done in the literature, such as High Energy Density Physics 9 (2013) 375]. In the present paper we present another way to modify the FFM to yield the correct profile in the (ion) impact limit and compare with the GBKO results for the static result.
-Lines 63-64: MD specifically refers to taking perturber-perturber interactions into account. GBKO does not do that and neither does FFM(there is no parameter I know of in FFM that is related to perturber-perturber interactions. I might add that there is no such parameter in the modification proposed by the authors either. For instance would it matter if you had 50% singly charged protons, 20% doubly charged Carbon ions and 30% triply charge nitrogen?)
So Rewrite the first paragraph of Section 2 as “ The main purpose of this work is to augment the FFM, because if the FFM can be robust and cover the ion impact regime without significant loss of speed, this would present an ultrafast method for lineshape calculations in a plasma. In the present paper we consider the linear and quadratic Stark effects separately.
-Lines 88-97 should read: In the case of the linear Stark effect, the FFM’s problems in the ion impact regime may be attributed to an incorrect dependence on the ion velocity[13]. The present work proposes a modification, namely replacing the constant jumping frequency of FFM with a nonconstant one from [4] that correctly interpolates between the static and impact limits. We then show by analytical and numerical calculations that this recovers the correct impact limit. For the quadratic Stark effect we compare with GBKO.
Comment: GBKO is WAY OLD! There are MUCH better calculations today. GBKO is a pertrbative theory that needs a ‘strong collision term’ for close collisions and this is an estimate that is quite uncertain and can be significant. I would also like to ask the authors the following question, that they may wish to answer: If we focus on the impact limit, then the perturbative expressions involve the energy spacings waa’, e.g. the energy differences between upper-upper perturbing levels. There is NO such parameter in FFM. There is also no parameter to account for penetrating collisions. There is no parameter to account for unitarity. This is why I am skeptical on the work here: Yes, you can get the correct functional (asymptotic) dependence, but this does not mean the result is correct. The MMM also has the right limits, but gives incorrect results. This is in contrast to other proposals to improve FFM as discussed above. I would like the authors to acknowledge and possibly discuss this.
Section 3:
-The first sentence is incomprehensible. Do the author really mean that the ‘complicated dynamics’ do not matter? To an extent they are washed out, for very long times, but you cannot really say that in general. The next sentence is also confusing. What “makes jump” is unclear as stated? The field or the intensity?
Second paragraph of Section 3:
-Leave out “great” . Molecular dynamics is typically abbreviated MD, not MMD
-Also, I would substitute “particle simulations” for MD, as the FFM does not account for perturber-perturber interactions, as MD does.
-Elaborate on “the theorem about equivalence of the FFM procedure to the solution of the kinetic equation with the so-called “strong collision integral”.
These should be defined. As it stands it is not immediately clear what kinetic equation and what strong collision integral the authors are talking about (typically the first thing that would come to mind is the strong collision term associated with the breakdown of perturbation theory”. However, ion impact is typically non-perturbative (see references above)
-Line 101 “constant of the Stark Effect” should read “Stark constant”
Section 4
First line of the Section “In the case of the Holtsmark theory, e.g. unshielded Coulomb interactions and no perturber-perturber interactions….”
-First line after Eq(18). I assume this is a ‘v’ as ‘velocity’ rather than ‘nu’ that goes to infinity. It looks like nu .
-The lines after 117 are the essence of this work and should be rephrased as follows:
“Our approach is to replace the constant jumping frequency nu of the FFM by a function f(nu,z) and use (14) for the lineshape of a single Stark component. This function f is extracted from [4] and designed to recover the static and impact limits, as follows:…
-Line after 124: Firstly should read First
-Line 126: After z^-5/2, add “Note that this asymptotic behavior is generic and does not depend on the assumptions of the Holtsmark field since strong fields are unshielded ones and the perturber-perturber correlations unimportant”.
-2 lines before Eq (27)
Nominater should read “nominator”. Same for denominator
-Line after 141 is unclear. Which of the profiles in Fig. 1 can be approximated by a Lorenzian? Certainly not the ones with the dip! And in the figure or the legend the 0.1, 1.0, 15.0 etc must be labelled, e.g. nubar
Also it is unclear to me how the authors used least squares. They are supposed to present 2 series of calculations, one with constant and the other with variable nu.
Also I do not see that in Fig.2 the constant and nonconstant results have widths with ‘completely different values’. One set has identical widths and the other (the wider one) slightly different as far as I can see from the graph. Am I misreading something or maybe there is a mislabel?
Fig.3 requires more explanation. If I understand correctly, the points are numerical and the dashes Eqs (31) and (32)? This should be stated explicitly, not just ‘curves’. Also, the authors should comment on the static limit.
-Lines 151-153. Like I said, I do not see that the authorss’ modifaction ‘yields the correct results for the low and high T’. At best it yields the right asymptotic behavior, though this was not clear for the static limit in Fig.3. Certainly sentec in lines 156-157 is unsubstantiated in my view.
Section 5
First line above Eq(35): statical should read static
Line 163: no applicable should read noT applicable
Line 177 : should read in figure, not in the figure.
Line 181 Firstly should read First
Line 183 than THE scalar one
Line 189 again denominator
Line 193 is not instead of doesn’t
Fig.4 blue line typo: Vector, not Verctor
I must say that I am missing the point on CT. CT is way obsolete. What is the authors’s point? I also do not see the point with GBKO. GBKO is a perturbative impact theory, used for electron impact. If you want ion impact, then it is not clear that perturbative works, because you may end up with a large and uncertain strong collision term
I would like to ask the authors to consider these points and revise accordingly. I think the idea merits publication, but the manuscript must be revised as I have indicated in my detailed comments.
Author Response
Concerning referee #1, the Guest Editor of this Special Issue, Professor Oks notified us that referee #1 was unsuitable and that therefore we do not have to respond to his report. Therefore, we do not respond to referee #1
Reviewer 2 Report
This article deals with an aspect of Stark's spectral line broadening theory. Namely, the authors modify the so-called Frequency Fluctuation Model (FFM) such that the modified model leads to a correct relationship between the line width and the mean perturber velocity, within the high speed limits. For this purposes they introduced dependence of electric field jumping frequency in FFM on the field strengths, especially it is needed for linear Stark-effect.
The verification of the proposed modification was carried out for a single Stark component, linear and quadratic Stark effect, i.e. hydrogen and non-hydrogen emitters. However, one could as well say that such tests simply relate to an atomic oscillator model whose oscillation frequency shift is proportional to the first or second power of the electric field strength.
The proposal itself is very interesting and will be useful for those researchers who want to use FFM to model specific shapes of spectral lines.
The work is legible, extensive and contains many details. In this context, a laconic justification of the eqs. (20-21), which is very important from the point of view of the aim of the paper, is somewhat surprising. The authors should consider extending the description of the form of eqs. (20-21), but it is not obligatory.
Author Response
We are grateful to the Referee for his work.
Formulas (20-21) are the result of complicated calculations. We introduced the additional reference Chandrasekhar, S. (1943). Stochastic problems in physics and astronomy. Reviews of modern physics,15(1), 1. for readers who want more details about stochastic problems and field distribution problems.
Reviewer 3 Report
The manuscript
“From the Vector to Scalar Perturbations Addition in the Stark Broadening Theory of Spectral Lines”
by Astapenko et al.
presents an important findings and analyses related to the Stark broadening theory and its approximations by means of the Frequency Fluctuation Method FFM. The authors demonstrated that for the linear Stark effect, the correct behavior of the spectral line width in the impact limit can be obtained employing a non-constant frequency fluctuation that has previously obtained by Chandrasekhar and van Neuman in the context of gravitational field statistics. This is an important result that remained for really solved until the discovery in 2013 by Stambulchik and Maron that a constant frequency in the FFM does not lead to the correct behavior in the impact limit of a hydrogen plasma.
The authors represent their results in a very elegant manner via the normalized statistical profile that permit scaling studies and generalize parameter analysis rather than presenting specific line broadening calculations. Finally the authors obtained analytical fits for a non-constant frequency fluctuation that is in rather good agreement with numerical calculations.
Their results are of great practical importance due to the need of fast calculations of numerous shapes of transitions for integrated simulations (e.g. opacity studies,…). For these purposes, the FFM has been proved to be an excellent approximation and respective references are well cited.
In addition, the authors studied also the non-linear Stark effect with respect to constant and non-constant frequency fluctuation. Their finding is that in this case, a constant fluctuation frequency provides better results with the reference data that have been taken from the historical reference of Griem, Baranger, Kolb and Oertel (so-called GBKO theory). Whether the GBKO theory is an exact reference or not can be discussed in length (and the authors are well aware of it): however, this is not the subject of the paper and can occupy studies for the next generations, so I do not comment on this further in particular as the GBKO provides sufficient precision for the current discussion of the asymptotic behavior of the FFM.
What remains is a philosophical question: does a representation of a non-constant frequency fluctuation exist, that likewise describes linear and non-linear Stark effect with correct asymptotic behavior ?
Or are we still in a pragmatic century: for linear Stark effect take non-constant frequency fluctuation as derived in the manuscript but for non-linear Stark effect, take a constant one ? I would be great if the authors could comment on this.
It would also be helpful for the general readership if the authors could add some discussion concerning (in the context of their findings)
- a) effects of strongly coupled plasmas and respective modifications of the electric field distribution,
- b) ion dynamics in FFM.
Some comments that should be corrected/commented before publication:
1) Ref. 5 is incorrect, it should be
I.I. Sobelman, L.A. Vainshtein, E.A. Yukov, Excitation of Atoms and Broadening of Spectral Lines, 2nd edition, Springer 1995
2) Concerning Figs. 1, 2, 5 and 6: the choice of color and style is not very helpful to compare the different theories, it would be better to assign different jumping frequencies to the different colors and change the style (solid and dashed) for constant and non-constant fluctuation frequency
3) it seems that the statement related to eq. (22) is correct for Gauss and Lorentz profiles, but might in general not be correct. It seems, that even the Voigt profile has not this asymptotic behavior (parameter “a” that is a combination of Lorentz and Gauss width)
4) line 153: probably more correct would be to say that for reduced fluctuation frequencies smaller than one constant and non-constant FFM are close to each other.
Concerning the related discussion, it would be helpful if the authors could provide some physical arguments here instead of pure “numerical observation”.
5) Line 163: not clear what this phrase means in the context of the discussion and that GKBO are just doing this (text above eq. 35))….. needs better explanation.
Some minor corrections:
6) there are several typos:
- line 24: Holtsmark
- line 52: …for high velocities
- line 56: suppress “it” at the end
- line 73:…computational resources
- line 88:…which is connected
- line 90: …in replacing the …
- line 99: …entirely numerical..
- line 117: …by introducing a non-constant…
- line 173: …effect the formula…
- line 174: …integrals. For this reason we do not provide…
- line 175:….for any ion temperature…
Author Response
We are very grateful to the Reviewer for his work and deep understanding of our paper.
It would also be helpful for the general readership if the authors could add some discussion concerning (in the context of their findings)
-
a) effects of strongly coupled plasmas and respective modifications of the electric field distribution,
-
b) ion dynamics in FFM.
In order to overcame the problems of non-ideal plasmas one have to compare the Weisskopf and Debye radii. In the present work we don’t consider effects of non-ideal plasma. Ion dynamics is treated in the framework of the FFM.
1) Ref. 5 is incorrect, it should be
I.I. Sobelman, L.A. Vainshtein, E.A. Yukov, Excitation of Atoms and Broadening of Spectral Lines, 2nd edition, Springer 1995
-The reference presented in the paper is correct. In the old Sobelman’s book there is more information on the considered problem. However, the book indicated by the Referee included a lot of materials from old monograph so we included it our references.
2) Concerning Figs. 1, 2, 5 and 6: the choice of color and style is not very helpful to compare the different theories, it would be better to assign different jumping frequencies to the different colors and change the style (solid and dashed) for constant and non-constant fluctuation frequency.
-We tried our best to make good figures. In order to demonstrate our results we have to put several curves on one figure. We followed the Referee’s advise and reworked 1st and 2nd figured. In our opinion 5th and 6th figures look as good as they can.
3) it seems that the statement related to eq. (22) is correct for Gauss and Lorentz profiles, but might in general not be correct. It seems, that even the Voigt profile has not this asymptotic behavior (parameter “a” that is a combination of Lorentz and Gauss width)
-If the profile is normalized and symmetric about the center than it follow this relation. In can be understood from simple considerations: integral over all allowed values from f(x) equals to 1. Most of f(x) values are located in (-a/2:a/2) interval, where a is width of profile. So this integral is proportional to a*f(0)~1.(we suppose that in zero point f(x) has maximum)
4) line 153: probably more correct would be to say that for reduced fluctuation frequencies smaller than one constant and non-constant FFM are close to each other.
-We corrected this sentence
5) Line 163: not clear what this phrase means in the context of the discussion and that GKBO are just doing this (text above eq. 35))….. needs better explanation.
-This sentence was rewritten.
6) there are several typos…
-We are grateful to the Referee for finding these typos!
Concerning the related discussion, it would be helpful if the authors could provide some physical arguments here instead of pure “numerical observation”.
-We are focusing on the theoretical part of the problem. Moreover we made references on Sobelman’s monograph and several papers which contain estimations of g and consider real physical situations.
What remains is a philosophical question: does a representation of a non-constant frequency fluctuation exist, that likewise describes linear and non-linear Stark effect with correct asymptotic behavior ?
Or are we still in a pragmatic century: for linear Stark effect take non-constant frequency fluctuation as derived in the manuscript but for non-linear Stark effect, take a constant one ? I would be great if the authors could comment on this.
-For the Linear Stark effect we believe that the modification of the FFM describes the broadening in the framework of complicated ion dynamics. Also according to the Chandrasekhar and van Neuman works the jumping frequency depends on field strength. For the quadratic Stark effect situation is more complicated. Direct calculation shows that in this case the FFM performs better with a constant jumping frequency.
